# An Overview of Advances in Rare Cancer Diagnosis and Treatment

**DOI:** 10.3390/ijms25021201

**Published:** 2024-01-18

**Authors:** Grania Christyani, Matthew Carswell, Sisi Qin, Wootae Kim

**Affiliations:** Department of Integrated Biomedical Science, Soonchunhyang Institute of Medi-Bio Science (SIMS), Soonchunhyang University, Cheonan 31151, Chungcheongnam-do, Republic of Korea; gchristyani4869@gmail.com (G.C.); matthewcarswell020518@gmail.com (M.C.)

**Keywords:** rare cancers, rare cancer diagnosis, rare cancer treatment

## Abstract

Cancer stands as the leading global cause of mortality, with rare cancer comprising 230 distinct subtypes characterized by infrequent incidence. Despite the inherent challenges in addressing the diagnosis and treatment of rare cancers due to their low occurrence rates, several biomedical breakthroughs have led to significant advancement in both areas. This review provides a comprehensive overview of state-of-the-art diagnostic techniques that encompass new-generation sequencing and multi-omics, coupled with the integration of artificial intelligence and machine learning, that have revolutionized rare cancer diagnosis. In addition, this review highlights the latest innovations in rare cancer therapeutic options, comprising immunotherapy, targeted therapy, transplantation, and drug combination therapy, that have undergone clinical trials and significantly contribute to the tumor remission and overall survival of rare cancer patients. In this review, we summarize recent breakthroughs and insights in the understanding of rare cancer pathophysiology, diagnosis, and therapeutic modalities, as well as the challenges faced in the development of rare cancer diagnosis data interpretation and drug development.

## 1. Introduction

Cancer, an affliction that has persisted throughout human history, has emerged as a leading global cause of mortality. Despite its pervasive presence in contemporary society, attention is increasingly turning towards the investigation of exceptionally rare cancer cases that have sporadically appeared in historical records. The burgeoning capabilities of modern medical biology technologies have now facilitated the systematic documentation, analysis, and treatment of these seldom-encountered malignancies. While these unique cancers represent a subset of diseases with historical precedence, our comprehension of these enigmatic conditions remains constrained, primarily due to their infrequent incidence [1].

It is important to know the distinction between rare cancers and rare diseases. Rare cancers, as defined by the National Cancer Institute, manifest in fewer than 15 cases per 100,000 individuals annually [2]. For example, rare cancers, such as Hodgkin lymphoma and meningiomas, have an extremely low incidence rate of 33.56 and 1.92 per one million per year, respectively. In Europe, rare cancers are defined as those with an incidence of fewer than 6 per 100,000 people per year [3]. This definition applies to cancers such as Kaposi sarcoma, which has an incidence rate of 0.34 per 100,000, and Ewing sarcoma, which has an incidence rate of 0.13. Conversely, rare diseases are defined as those that affect fewer than 5 in 10,000 people. Currently, it is estimated that there are 6000 to 7000 different rare diseases. Although considered rare, together, they impact around 30 million individuals in the European Union [4].

Currently, there are 230 distinct rare cancers, including widely known examples such as Merkel cell carcinoma, thymic carcinoma, glioblastoma multiforme, hepatoblastoma, Ewing sarcoma, Kaposi’s sarcoma, esophageal cancer, chronic myeloid leukemia, acute lymphoblastic leukemia, and anal cancer, among others, each exhibiting diverse variations (see Supplemental Appendix A). In contrast to common cancers, the inconspicuous nature of rare cancers poses challenges for conventional diagnostic methods like biopsy and X-ray imaging, resulting in a scarcity of medical and research information on their characteristics. However, leveraging modern technologies enables researchers and medical practitioners to detect and study these elusive cancers, advancing diagnosis, treatment, and prognosis [5]. State-of-the-art diagnostic techniques encompass new-generation sequencing methods, RNA-sequencing, and omics, alongside the integration of artificial intelligence and machine learning for comprehensive analyses and functional studies of rare cancers [6]. The analysis of diverse data pertaining to the identification of rare cancers holds the potential for the development of innovative treatments. Although the therapeutic landscape for rare cancers is nascent, ongoing progress is evident, with treatments such as chemotherapy, surgery, transplantation, targeted therapy, and combination therapy being meticulously researched. Uncovering numerous oncogenic genes and rare cancer pathways has catalyzed the development of anti-tumor drugs, some of which have shown promise in clinical trials for treating rare cancers [7].

This comprehensive review endeavors to elucidate recent strides in the diagnosis and treatment of rare cancers within the current landscape of medical technology. By synthesizing recent breakthroughs in the understanding of rare cancer pathophysiology, diagnosis, and therapeutic modalities, we aim to furnish a compendium of groundbreaking insights. A graphical abstract of this review is provided.

## 2. Rare Cancers

Cancer presents formidable medical challenges encompassing aspects of prevention, prognosis, diagnosis, and treatment. Despite decades of established strategies and preventive measures, the inevitability of cancer persists, influenced by environmental factors, lifestyle choices, stochastic genetic mutations, and other variables. This inevitability extends to rare cancers, which may manifest concurrently and sporadically given sufficient time and optimal conditions for profound genetic mutations [8]. The intricate nature of cancer, coupled with its potential to elude detection even in the presence of copious scientific data and studies on prognosis and diagnosis, poses a significant obstacle. This challenge is exacerbated in the context of rare cancers, where limited research has been conducted on their prognosis, diagnosis, symptoms, and characteristics. The scarcity of studies and case reports, along with a lack of empirical evidence, hinders scientific comprehension and research. This scarcity also creates a substantial barrier to the identification and investigation of these conditions. Consequently, the development of effective cures or the selection of appropriate therapies proves to be a formidable challenge. Many rare cancers remain inadequately treatable due to these limitations. Compounding the predicament is the frequent conflation of diagnoses of rare cancers with more common forms, as they often exhibit similar symptomatic profiles. Incorrect diagnoses further complicate the situation, contributing to delayed detection, particularly given that rare cancers are prone to being asymptomatic at times. Late-stage detection is a common outcome, exacerbated by the elusive nature of symptoms associated with rare cancers [9].

Despite these challenges, a concerted global effort spanning several decades has been dedicated to addressing this issue, resulting in numerous breakthroughs. Advances in research strategies and technologies have provided unprecedented insights into these enigmatic cancers. These recent developments hold the promise of more precise diagnoses, improved treatment modalities, enhanced prognostic capabilities, and significant revelations about the pathogenesis of these diseases. In the context of these pioneering advancements, contemporary methodologies, including genome sequencing, RNA-sequencing, and omics technologies, among others, have been devised and are undergoing continuous refinement to augment their efficacy, precision, and cost-effectiveness. This concerted effort aims to enhance the comprehension of rare cancers. The amalgamation of these technologies within the realm of applied science holds the potential to facilitate more sophisticated scientific revelations, thereby enabling early diagnosis and detection of rare cancers, and mitigating their malignant progression.

## 3. Rare Cancer Diagnosis

### 3.1. Conventional Diagnosis

Inherently, the diagnostic procedures for rare cancers parallel those employed for conventional cancers. These diagnostic modalities encompass a comprehensive array of techniques commonly utilized in the examination of prevalent malignancies, including physical examination techniques such as pressure application, histological studies, fine needle aspiration cytology (FNAC), biopsy, X-ray imaging, and electron microscopy. Additionally, molecular and chemical analyses play a pivotal role, incorporating histo-/cytochemistry, detection of tumor markers, flow cytometry, in situ hybridization, and DNA microarray techniques.

Despite the effectiveness of these conventional diagnostic approaches in common cancers, their application encounters inherent limitations pertaining to accuracy and early detection in rare cancers. The resemblance in physical characteristics between common and rare cancers often results in the inadequate and inaccurate diagnosis of the latter. Notably, molecular and chemical analyses emerge as more efficacious in facilitating the early detection of rare tumor progression. A comprehensive overview of these conventional diagnostic techniques for cancer is presented in Table 1.

### 3.2. Modern Diagnosis

Modern diagnosis utilizing novel techniques has been playing an important role in the development of rare cancer diagnoses (see Figure 1).

#### 3.2.1. Genome Sequencing

The utilization of genome sequencing, particularly DNA sequencing, traces its origins back to 1975 with the establishment of the first DNA sequencing system by Sanger [16]. Employing the sequencing-by-synthesis approach, this method involved radioactively labeled DNA strands complementary to the template strand, utilizing the di-deoxy chain termination technique. Subsequent advancements, automation, and commercial availability marked the evolution of this technique, referred to as first-generation sequencing (1st GS) [17]. Concurrently, discoveries such as cDNA PCR technology by Iscove [18] and reverse transcriptase [19] catalyzed a revolutionary transformation in genome sequencing, culminating in the initiation of the Human Genome Project in 1990. By mid-2003, this monumental scientific endeavor had successfully identified the entire human DNA, providing fundamental insights into the human blueprint and significantly advancing the study of human biology and medical practices.

Despite its groundbreaking contributions, 1st GS had limitations in terms of throughput and cost, generating only one sequence per electrophoresis lane or capillary tube. Addressing these challenges, Next/Second-Generation Sequencing (NGS) emerged as a high-throughput, massively parallel sequencing approach. Unlike 1st GS, NGS operates without segregating sequencing reactions into lanes, capillaries, or tubes, enabling billions of simultaneous sequencing processes on a slide surface (glass or beads). This remarkable improvement in throughput and cost facilitated the widespread implementation of NGS instruments from various manufacturers, including Illumina Ion Torrent, Qiagen, Genapsys, and Roche [20,21,22]. NGS technology introduced innovative sequencing techniques such as whole genome sequencing (WGS), whole exome sequencing, RNA-sequencing, and omics, among others.

Contemporary applications of these technologies encompass the detection of cancer-causing mutations, identification of specific cancer biomarkers, and investigation into the underlying mechanisms of mutations, including their causes, effects, and potential treatments. Noteworthy studies utilizing WGS include Wang et al.’s [23] identification of somatic variant loci associated with hepatoblastoma, revealing deleterious mutations in Ctnnb1, Axin2, and Parp1. Additionally, Pagnamenta et al. [24] identified novel oncogenic genes, and Chojnacka et al. [25] explored rare structural variants in multiple myeloma. Moreover, Lee et al. [26] employed WGS to diagnose a rare SDHB-deficient metastatic renal cell carcinoma, exemplifying the capability of WGS in the precise and early diagnosis of rare cancers through the scrutiny of genetic mutations and markers of oncogenic genes. Furthermore, WGS can be utilized in the clinical diagnosis of cancer or its rare variants. For example, Turro et al. successfully detected cancer-related mutations in peripheral blood from participants with a personal history of tumor-associated phenotypes (identified through 11 Australian Familial Cancer Centers), which was used to measure their susceptibility to cancer and predict cancer type [27]. In summary, genome sequencing has played a pivotal role in advancing our understanding of rare cancer genetics by providing a detailed and personalized view of the genomic landscape. This information has far-reaching implications for diagnosis, treatment, and ongoing research efforts in the field of rare cancers.

#### 3.2.2. RNA-Sequencing Analysis

Rare cancers may exhibit unique genetic alterations, such as fusion genes or alternative splicing events, driving oncogenesis. Therefore, using RNA-Seq can be advantageous for unraveling these complex molecular signatures as it facilitates quantitative measurement of gene expression levels, essential for understanding alterations in specific genes in rare cancer cases compared to normal tissues. Through gene expression profile comparison, the dysregulated biological processes and pathways in rare cancers can be better understood. With a copious amount of poorly understood rare cancers, the discoveries of novel transcripts and isoforms specific to rare cancer subtypes would be inevitable and thus may unveil previously unknown biomarkers or therapeutic targets. RNA-seq’s ability to detect specific RNA molecules, including mRNA, non-coding RNA, and splice variants, has been used as a diagnostic or prognostic biomarker for rare cancers. These biomarkers differentially express and correlate with clinical outcomes, facilitating the development of accurate diagnostic tools and prognostic indicators. RNA-seq has been extensively studied and applied to the diagnosis of rare cancer. For example, rare cancers such as gliomatosis peritonei can be detected by identifying high expression of the stem cell marker Sox2 and low expression of the transcription factors Oct4 and Nanog [28]. Moreover, Pei et al. reported an accurate diagnosis by identifying disease-specific fusion genes in cancer patients from the Department of Pathology at Fox Chase Cancer Center [29]. They successfully diagnosed an Ewing sarcoma by analyzing cancer tissue samples using RNA-seq analysis.

The most recent iteration of RNA-seq allows for the sequencing of individual cells as opposed to the conventional approach, which involves sequencing thousands of cells simultaneously. Various techniques are presently employed to isolate individual cells, such as manual cell selection, limiting dilution, laser-capture microdissection [30], fluorescence-activated cell sorting [31], magnetic-activated cell sorting [32], and microfluidics [33]. Microfluidics, particularly droplet-based microfluidics (also known as microdroplets), has gained popularity due to its minimal sample requirements, precise fluid control, and cost-effectiveness [34]. In microdroplets, single cells are encapsulated within nanoliter droplets containing a lysis buffer and barcoded beads, achieved through microfluidic and reverse emulsion devices [35]. Another RNA-seq variant, such as massively parallel single-cell RNA sequencing, represents an automated, upgraded version of cell expression by linear amplification and sequencing. This method can investigate cellular heterogeneity within the immune system by establishing an automated experimental platform for RNA profiling of cells sorted from tissues using flow cytometry [36,37]. Overall, RNA-seq has been proven to be useful in rare cancer diagnosis, especially with the current advancement in molecular technology, and this method can complement other diagnostic techniques to further increase their precision in cancer detection.

#### 3.2.3. Omics Analysis

Omics is defined as analyzing large amounts of data that contain information about the structure and function of an entire biological system at a particular level. This new emerging technology includes genomics, transcriptomics, proteomics, metabolomics, epigenomic, epi-transcriptomic, epi-proteomics, DNA-RNA interactomics, RNA-RNA interactomics, DNA–protein interactomics, RNA–protein interactomics, protein–protein interactomics, and protein–metabolite interactomics. Omics play a crucial role in enhancing cancer diagnosis. These high-throughput approaches provide a comprehensive view of the molecular landscape of cancer, offering valuable insights into the genetic, transcriptional, protein, and metabolic alterations associated with the disease. Genomics includes the DNA microarrays, 1st GS, 2nd GS, and 3rd GS, which have been discussed earlier. This type of omics also includes comparative genomic hybridization (CGH) and Single Nucleotide Polymorphism (SNP) arrays to help detect amplifications or deletions of specific genomic regions, providing information about chromosomal instability in cancer cells. Meanwhile, transcriptomic analysis reveals the expression levels of genes in cancer cells, helping to classify tumors, predict their behavior, and identify potential biomarkers for diagnosis. This analysis can uncover alternative splicing events that may lead to the production of variant proteins associated with cancer progression.

Proteomic techniques, such as mass spectrometry and protein microarrays, provide information about the expression levels of proteins in cancer cells. Abnormal protein expression can serve as diagnostic markers or therapeutic targets. The abnormality includes post-translational modifications such as phosphorylation and acetylation, which play a role in cancer signaling pathways. On the other hand, metabolomics examines the small-molecule metabolites in cells, tissues, or biofluids. Altered metabolic profiles in cancer cells can be indicative of specific metabolic pathway dysregulation and provide insights into the tumor microenvironment. Moreover, metabolomic analysis may identify metabolic biomarkers that can aid in cancer diagnosis, prognosis, and treatment response monitoring.

By combining these methods, multi-omics integration can be achieved. This combined data is integrated from multiple omics layers (genomics, transcriptomics, proteomics, metabolomics, and other omics) to provide a more comprehensive understanding of the complex molecular networks underlying cancer.

Another use of omics technology is its ability to detect circulating tumor DNA. Genomic analysis of circulating tumor DNA provides non-invasive monitoring of genetic alterations in tumors that lead to early cancer detection, monitoring treatment responses, and detecting minimal residual disease. Lastly, with the surging capabilities of AI and machine learning algorithms, these technologies can analyze large omics datasets to identify patterns, classify tumors, and predict patient outcomes, enhancing the diagnostic accuracy and precision of personalized medicine approaches.

Although the integration of omics in rare cancer diagnosis is still limited, their capability to accurately detect and analyze cancer biomarkers is a proof of concept. Currently, this technique is only used to support research on rare cancer diagnoses, such as the identification of biomarkers. However, incorporating omics technologies into cancer diagnosis will contribute to a more detailed and personalized understanding of the disease, paving the way for targeted therapies, early detection, and improved patient outcomes. As technology continues to advance, omics approaches are likely to play an increasingly significant role in the field of oncology.

#### 3.2.4. Functional Studies Analysis

Functional studies provide rare cancer diagnosis with insights into such cancer behavior and characteristics. These studies determine the functional aspects of rare cancer, such as its growth patterns, response to treatments, and the underlying molecular and cellular mechanisms. This diagnosis tool takes advantage of several different technologies, including immunohistochemistry (IHC), flow cytometry, fluorescence in situ hybridization (FISH), functional magnetic resonance imaging (fMRI), cell culture studies, and functional genomics.

IHC, as previously discussed in Table 1, is a technique that detects specific proteins inside a rare cancer by using antibodies. Those biomarkers include hormone receptors, proliferation markers, and specific oncogenes or tumor suppressor genes [38]. Rare cancer biomarkers such as Pax8, Pax2, napsinA, carbonic anhydrase IX, claudin-4, Cdx-2, and others have been identified and studied thoroughly to help provide a better diagnosis of rare cancer [30].

However, IHC has a big limitation in its inability to tag more than one marker per tissue section, besides its dependency on examiner interpretation of data. To tackle this issue, multiple IHC variations have been made and continue to be developed to satisfy current clinical needs. The introduction of multiplex IHC (mIHC) is one of them. This alternative IHC method uses tyramine chemistry to analyze multiple biomarkers at the same time. It uses a wide range of new chromogenic dyes, hence making in situ analysis with standard brightfield microscopes possible. These dyes can be used individually or mixed to create new colors, which can provide signals similar to the conventional 3,3′-diaminobenzidine chromogen. They also allow the examination of co-localized biomarkers. The chromogens have broad absorbance spectra, which result in distinct dark staining patterns that can be easily distinguishable under light microscopy [39]. Importantly, standard scanners can also capture images of these stained slides, aiding biomarker research and the potential development of in vitro diagnostic products.

There are alternative methods for chromogenic mIHC, one of which is multiplexed immunohistochemical consecutive staining on a single slide. In this method, iterative cycles involve tagging, image scanning, and destaining of a chromogenic substrate—all performed on a single slide. This innovative technology captures the intricacies of the immunome, facilitating high-dimensional immunohistochemical analyses within the context of routine pathology workflow and standards. It’s noteworthy that the effectiveness and applicability of chromogenic mIHC/immunofluorescence (IF) in research settings have been observed, especially when paired with color unmixing [40] using advanced algorithms, ultimately resulting in improved accuracy during image analysis.

Another IHC variant is metal-based mIHC/IF, which encompasses techniques such as imaging mass cytometry (IMC), multiplexed ion beam imaging (MIBI), and fluorescence-based mIHC/IF. IMC employs high-resolution laser ablation and mass cytometry to simultaneously assess over 100 biomarkers (though practically > 40 due to isotope availability) from tissue sections labeled with metal-tagged antibodies [41]. The main advantage is the extensive marker examination, but challenges include the need for thorough antibody validation and limitations in acquisition speed, constraining the imaged area. To address this, immunofluorescence is employed to selectively analyze specific regions of a slide in IMC analysis.

MIBI is a novel approach involving the staining of tissues with 40–100 metal-labeled antibodies. In MIBI, tissues are ionized by high-energy beams, generating secondary ions detected by an imaging mass spectrometer across a five-log dynamic range [42]. Another recent multiplex technology is TSA-based mIHC/IF, like Vectra. This method allows simultaneous antibody-based detection and quantification of up to six protein markers (recently extended to nine or more markers) on a single tissue section, along with a nuclear counterstain [43]. Vectra has gained widespread adoption in the past half-decade, with several notable publications globally. It holds the potential for refining diagnostic criteria and assessing predictive biomarker values, particularly in lymphoid pathology, offering reproducibility and reliability [44]. Some institutions and hospitals use the Vectra system in clinical laboratory tests to assist clinicians in decision-making and treatment plans.

Utilizing a strategic approach, chip-cytometry functions as a versatile platform technology aimed at elevating data quality across the entirety of the analysis pipeline, encompassing staining, imaging, and the final analytical phase. Within a microfluidic chip, tissue sections, whether conserved as formalin-fixed, paraffin-embedded, or fresh-frozen, find placement on adherence-enhanced coverslips, forming a sealed chamber. This arrangement facilitates multiple staining processes, including iterative cycles of staining, imaging, and bleaching/quenching. This enables the repetitive application of up to five colors, generating a substantial array of markers.

Digital spatial profiling (DSP), on the other hand, emerges as a technique for high-plex spatial profiling of proteins and RNA, harnessing oligonucleotide detection technologies. In DSP, oligonucleotide barcodes link to antibodies via a photocleavable UV light-sensitive linker. The use of UV light allows for the detachment of high-plex oligo tags from the antibody, retrieved from the tissue surface, thereby enabling sample reuse. Quantitative analysis of the oligo barcodes ensues, followed by mapping back to tissue locations for spatial profiling at specified regions of interest. DSP distinguishes itself as a high-throughput technology, with the capability to analyze 16–20 sections daily [45]. Successful DSP applications involve the characterization of the tumor profile in melanoma patients undergoing checkpoint blockade therapy, uncovering a correlation between baseline immune infiltration and treatment response [46]. Validation studies emphasize the robust detection of high-abundance protein and RNA targets [47]. When compared to IHC, DSP showcases a significantly wider dynamic range and demonstrates high concordance with quantitative immunofluorescence, validated through regression and outcome assessment [48].

Flow cytometry, as defined in Table 1, adds additional data to functional studies analyzing the physical and chemical characteristics of cells in suspension. In cancer diagnosis, flow cytometry can be employed to assess the DNA content, cell cycle distribution, and expression of surface markers in cancer cells. This information helps classify tumors and predict their behavior. An example of this technique is imaging in vivo flow cytometry, which can detect circulating cells at concentrations as low as 20 cells per mL and could reach 50% sensitivity in conditions with two orders of magnitude-degraded contrast [49]. The latest variant of FC is full-spectrum flow cytometry, which characterizes the physical and fluorescent properties of cells in suspension by using fluorochrome-conjugated antibodies to measure proteins expressed by distinct immune cell subpopulations [12]. FSFC was used in a study to validate CD38 expression in macrophages, using CD14, CD16, CD11c, CD68, CD11b, and HLA-DR for myeloid lineage and cell subset definition [50].

In addition to that, FISH is also used as supporting data for functional studies. FISH is a molecular cytogenetic technique that detects and locates the presence or absence of specific DNA sequences. It is particularly useful for identifying chromosomal abnormalities, gene amplifications, and translocations associated with certain types of cancer, aiding in the diagnosis and prognosis of the disease. Meanwhile, fMRI, which is a non-invasive imaging technique that measures and maps brain activity by detecting changes in blood flow, can be used for the early detection of cancerous lesions deep inside the body for a better understanding of cancer’s position and its structural characteristics. fMRI can be applied to study tumor vasculature, assess blood perfusion, and evaluate treatment response. It is often used in brain cancer diagnosis and treatment planning. Lastly, PET helps visualize metabolic activity in tissues, allowing for the identification of cancerous lesions, assessment of tumor aggressiveness, and monitoring of treatment response. PET involves injecting a small amount of radioactive material into the body, which is then detected by a PET scanner.

These functional studies, when used in combination, contribute to a comprehensive understanding of cancer biology and aid in personalized diagnosis and treatment planning for patients. Integrating multiple approaches allows clinicians to gather information on various aspects of cancer function, helping to tailor therapeutic interventions for better outcomes.

#### 3.2.5. Integrated Analysis

The integrated approach to cancer diagnosis involves amalgamating information from diverse sources and methodologies, aiming to achieve a more exhaustive and precise comprehension of the disease. This multifaceted strategy enables clinicians and researchers to concurrently consider various aspects of cancer biology. The integrated analysis encompasses a range of rare cancer data, incorporating genomic analysis, transcriptomic analysis, proteomic analysis, epigenomic analysis, imaging techniques, clinical and pathological data, and machine learning and bioinformatics.

Genomic analysis, comprising NGS, Comparative Genomic Hybridization (CGH), and Single Nucleotide Polymorphism (SNP) arrays, has been integral to studying and diagnosing rare cancers for decades. NGS technologies permit the sequencing of entire genomes or specific gene panels, offering insights into genetic mutations, alterations, and variations linked to cancer. Meanwhile, CGH facilitates the identification of chromosomal copy number variations, amplifications, and deletions in cancer genomes. SNP arrays are adept at discerning subtle genetic variations and allelic imbalances associated with cancer.

Transcriptomic analysis, as exemplified by microarray analysis, allows for the measurement of gene expression levels, aiding in the identification of differentially expressed genes and signaling pathways relevant to cancer. Conversely, RNA-Seq provides a high-throughput method for scrutinizing the transcriptome, unveiling information about gene expression, alternative splicing, and fusion genes.

Another facet of integrated analysis is proteomic analysis, which encompasses mass spectrometry and protein microarrays. Mass spectrometry facilitates the identification and quantification of proteins, enabling the exploration of protein expression, post-translational modifications, and protein–protein interactions. Concurrently, protein microarrays assess the expression levels of multiple proteins simultaneously, shedding light on protein function and signaling pathways.

Epigenomic analysis includes DNA methylation profiling and histone modification analysis. DNA methylation profiling examines DNA methylation patterns, aiding in the identification of epigenetic changes associated with cancer and contributing to the understanding of gene regulation. Histone modification analysis studies modifications to histone proteins, providing insights into chromatin structure and gene expression regulation.

Medical imaging (MRI, CT, PET) and radiomics play a pivotal role in integrated studies of rare cancers. Integrating imaging data with molecular information enhances tumor characterization, assesses size and location, and monitors treatment response. Radiomics further amplifies the data by analyzing quantitative features extracted from medical images, allowing for a more detailed and personalized assessment of tumor characteristics.

Clinical and pathological data, including electronic health records and pathological evaluation, are integral components of integrated analyses. Combining clinical data, encompassing patient history, treatment responses, and outcomes, helps correlate molecular findings with clinical observations. Additionally, integrating traditional histopathological analysis with molecular data provides a comprehensive understanding of tumor morphology and behavior.

Lastly, advanced computational methods and bioinformatics tools are indispensable for integrating and analyzing large-scale omics data. Machine learning technologies can create models trained to identify patterns, predict outcomes, and aid decision-making based on integrated data.

A summary of currently used diagnostic methods and technologies is presented in Table 2.

## 4. Rare Cancer Treatment

Finding new treatments for rare cancers is a challenge due to their small sample populations and the absence of a standardized treatment regimen, which can lead to further problems, including delayed diagnosis as well as economic challenges. The challenges and perspectives faced in finding novel rare cancer treatments are summarized in Table 3.

Despite the challenges and the need for novel rare cancer treatments, most healthcare institutions still treat rare cancer patients with conventional cancer therapy that is used for common cancers. In this section, we discuss rare cancer treatment modalities that can be divided into conventional therapy and modern therapy. Surgery, chemotherapy, and radiation are classified as conventional therapies for rare cancer, while modern therapy includes immunotherapy, targeted therapy, transplantation, and drug combinations. Modern therapy includes novel treatments that are used to mitigate the challenges of rare cancer treatment development. The examples of these rare cancer therapies are summarized in Table 4.

### 4.1. Conventional Therapy

The conventional therapy for cancer that is used as first-line therapy in common cancer treatments includes surgery, chemotherapy, and radiation therapy as adjuvant or neoadjuvant therapy. Without the presence of a rare cancer-specific treatment, conventional therapy for common cancers is still used as the initial treatment for rare cancers. As the mainstay treatment, surgery is often used as a primary method to manage solid tumor malignancies. With the development of surgical technology, invasive surgical techniques that are used to treat common cancers are replaced with novel surgical techniques that are minimally invasive and have higher safety and efficacy. The novel surgical techniques that are applied to rare cancer patients include thymectomy through video-assisted thoracoscopic surgery (VATS) and robotic VATS for thymic carcinoma, robot-assisted hepatectomy for hepatoblastoma, fluorescence-guided surgery for glioblastoma multiforme, and minimally invasive esophagectomy for esophageal cancer. Anal cancer, on the other hand, is usually treated by chemoradiation and chemo drugs, while abdominoperineal resection is used as an alternative modality when the main treatment is ineffective or when the anal cancer relapses.

Radiation therapy is used for rare cancer treatments, not only for adjuvant and neoadjuvant therapy but also when the cancer is found to be unresectable and locally advanced, to ensure precise targeting of the tumor and minimize the radiation exposure to the surrounding healthy tissues. Stereotactic body radiation therapy (SBRT) has been found to be effective in thymic carcinoma, with a 96.9% response rate and 62.2% median tumor shrinkage rate after 56 Gy SBRT. However, the combination treatment of three fractions of 24 Gy SBRT and an immune checkpoint inhibitor failed to improve efficacy in Merkel cell carcinoma [70]. Advances in radiation therapy for glioblastoma include the dose escalation of image-guided radiation therapy. Specifically, the standard treatment was 60 Gy in 30 fractions or 40 Gy in 15 fractions, which has been escalated to 72–76 Gy in 30 fractions with acceptable toxicity for tumor ˂ 100 cm^3^. Another type of radiation technique, selective internal radiation therapy, is the radiation treatment used for hepatocellular malignancy, mainly hepatocellular carcinoma and liver metastases, yet there are limited clinical data regarding the use of selective internal radiation therapy in pediatric hepatoblastoma.

Chemotherapy is commonly used as adjuvant therapy to ascertain the removal of tumors after surgery. To date, the chemotherapy used for rare cancer treatment is mostly similar to that used for common cancers, although most of the chemotherapies used for rare cancers are usually a combination of several types of chemo drugs. Chemotherapy drugs include alkylating agents (temozolomide, cyclophosphamide), antimetabolites (e.g., pemetrexed, fludarabine, fluorouracil) [73], antitumor antibiotics (e.g., danunorubicin, doxorubicin) [75], plant alkaloids (e.g., paclitaxel, vincristine) [73,74], topoisomerase inhibitors (e.g., Etoposide) [71], platinum compounds (e.g., cisplatin) [71], and hypomethylating agents (e.g., 5-azacytidin) [76]. Despite being a cornerstone treatment for rare cancers, the chemotherapy regimen also comes with a lot of drawbacks, as it is nonspecific and cytotoxic to any rapidly proliferating cells in the body, including hair, bone marrow, and the gastrointestinal tract. The limitations of chemotherapy also include drug resistance, dosage selection difficulties, and rapid drug metabolism.

### 4.2. Modern Therapy

#### 4.2.1. Immunotherapy

One of cancer’s emerging hallmarks is its ability to avoid immune destruction. Immunotherapy is a type of cancer modality that works by inducing the body’s immune response against cancer and plays a key role in suppressing cancer as it is the key to cancer cell elimination. Moreover, immunotherapy is the pivotal factor in inhibiting the metastasis of cancer, which favors the immunocompetent host microenvironment. Similar to the treatment for common cancers, the immunotherapy used for rare cancers can be classified into several types, as each type of immunotherapy has a different mechanism of action, targeted antigen, immune checkpoint target, and is often tailored to specific cancer types.

##### Immune Checkpoint Inhibitors (ICIs)

ICI is a type of cancer immunotherapy commonly used for rare cancer treatments. ICI blocks specific proteins on the surface of T-cells and cancer cells. The most commonly used ICIs are the inhibitors of Programmed Cell Death Protein 1 (PD-1) and its ligand (PD-L1). PD-L1 presents on cancer cells, binds to PD-1 on T-cells, and allows T-cells to eliminate tumor cells [105].

The PD-1/PD-L1 inhibitor has been studied in thymic epithelial tumors, which include thymic carcinoma (thymic epithelial tumor type C), due to its high expression level of PDL-1 [78]. Nivolumab (Opdivo) and Pembrolizumab (Keytruda) are PD-1 inhibitors that are commonly used to treat several types of rare cancers. Kaposi’s sarcoma patients treated with intravenous administration of pembrolizumab (200 mg; q3 weeks for 6 months) showed a 71% overall response rate (ORR) [85]. PD-1/PD-L1 inhibitor has also become the standard of care for advanced Merkel cell carcinoma [51]. A randomized, placebo-controlled Phase 3 Clinical Trial KEYNOTE-590 has reported that compared to chemotherapy alone, pembrolizumab plus chemotherapy has a superior effect in oesophageal cancer treatment with 13.9 months of median overall survival (OS) and 6.3 months of progression-free survival (PFS) [79]. On the other hand, although nivolumab is currently being studied for treatments in patients with glioblastoma, nivolumab usage has failed to show a significant survival benefit [106]. In anal cancer, the Phase II clinical trial of a PD-1 antibody, retifanlimab (INCMGA00012), presented an acceptable safety profile with 13.8% ORR and 10.1 months of median OS [88].

Though ICIs have a profound impact on the development of rare cancer treatments, they benefit a limited portion of cancer patients due to the immune-related adverse events (irAEs) during ICI therapy, which weaken the body’s natural defenses against autoimmunity and lead to diverse autoimmune reactions both locally and systemically [85]. Interestingly, the emergence of irAEs appears to be correlated with improved therapeutic efficacy. Phase 2 clinical trial of pembrolizumab in Kaposi’s sarcoma showed 76% of treatment-related adverse events, yet 12% of patients showed complete response and 59% showed partial response [78]. To reduce irAEs, ongoing studies are aiming to discover essential predictive biomarkers that can prognosticate patients who will respond positively to treatment and who are likely to develop irAEs [85]. In advanced melanoma patients who developed irAEs after anti-CLTA-4 and anti-PD1 treatment, mutations were enriched in seven genes [107]. Further genetic analysis could offer valuable information for predicting these adverse effects and guiding treatment options in rare cancer immune therapy.

##### Cytokines

Cytokines play a significant role in cancer immunotherapy as a signaling molecule that induces an immune system response to cancer cells. Interferon-α (IFN-α) was the treatment used for Chronic Myeloid Leukemia (CML) patients before the emergence of Tyrosine Kinase Inhibitor (TKI) as the standard treatment for CML. However, a high number of relapses has become a problem for rare cancer patients treated with TKI, as TKI is unable to eradicate CML progenitor cells, unlike IFN-α, which targets CML stem cells [108]. Therefore, there is potential for IFN-α to re-emerge as a CML treatment as a combination therapy with TKI to lower the chance of relapse or resistance of TKI during CML therapy. In glioblastoma multiforme, the cytokine profile is used as an important indicator to determine glioblastoma progression, as a short burst of inflammatory cytokines can suppress tumor progression, yet on the other hand, chronic inflammation leads to lower immune ability against glioblastoma [109]. In addition, pro-inflammatory cytokines IL-1β, IL-6, IL-8, and IL-10 and anti-inflammatory cytokines TNFα and HMGB1 are shown to play roles in the interaction of neutrophil and glioma cells [109].

##### Cancer Vaccines

Cancer vaccines are also a type of immunotherapy used in the rare cancer regimen. Polyinosinic-polycytidylic acid with poly-L-lysine and carboxymethylcellulose (Poly-ICLC) is a Toll-like receptor 3 agonist that is used as a type of cancer vaccine that works by mimicking the presence of viral RNA to stimulate the innate immune system and cytokine production. Poly-ICLC combined with tremelimumab, a Cytotoxic T-Lymphocyte-Associated Protein 4 (CTLA-4) inhibitor, and durvalumab, a PD-L1 inhibitor, was studied in Phase 1/2 clinical trials (NCT02643303) against Merkel cell carcinoma. Combined with tremelimumab and durvalumab, poly-ICLC was predicted to create in situ vaccination, modify the tumor microenvironment, and thus trigger the systemic immune response against cancer [77]. However, the study results showed no significant improvement in the OS of Merkel cell carcinoma patients treated with this combination.

Vigil is a personalized cancer vaccine that utilizes autologous tumor tissue transfected with granulocyte-macrophage colony-stimulating factor and bi-directional short hairpin RNA targeting the furin gene DNA to enhance the immunogenicity of tumor cells. Phase 1 clinical trial of Vigil treatment in Ewing’s sarcoma compared to control showed a significant increase in actual survival (73% vs. 23%) and median OS (731 days vs. 207 days) [84].

Another cancer vaccine, Axalimogene Filolisbac (ADXS11-001), has shown beneficial effects in cervical cancer; however, when it was used to treat squamous cell carcinoma of the anorectal canal, it only reached 3.4% ORR and 15.5% of the 6-month PFS rate, even though it has satisfactory treatment safety [110]. Thus, the use of ADXS11-001 might need to be combined with other treatments to achieve a significant therapeutic effect in anal cancer patients.

##### Monoclonal Antibodies

A monoclonal antibody is an antibody derived from a single cell and can only bind to one specific antigen. Since it is able to bind specifically, different types of monoclonal antibodies are being developed for treatment against rare cancers. Cetuximab and Bevacizumab are monoclonal antibodies that target Vascular Endothelial Growth Factor (VEGF) and Epidermal Growth Factor (EGF), respectively. Although the efficacy of the Cetuximab and Bevacizumab combination must be further improved, the Phase 1 clinical trial showed that their combination can be safely administered intraarterially to glioblastoma patients [80].

ABT-414 (Depatuxizumab mafodotin) is composed of an epidermal growth factor receptor (EGFR) monoclonal antibody, depatuxizumab, and mafodotin [81]. EGFR amplification can produce a unique EGFR protein conformation and expose the depatuxizumab binding site specifically in tumor cells, thereby allowing ABT-414 to bind to tumor cells that overexpress EGFR, allowing depatuxizumab to enter the cells and cause cell death. Since depatuxizumab has low binding with normal cell EGFR, it reduces the toxicity possibility. It is consistent with the Phase 1 clinical study in glioblastoma patients that shows reduced toxicity with ABT-414 treatment with 30.8% of PFS at 6 months and 10.7 months of median OS [81].

In B-ALL patients, blinatumomab, a bispecific antibody for CD3 and CD19 T-cells is reported to have high efficacy with a two-year OS compared to the chemotherapy group (71.3% vs. 58.4%) with a low toxicity profile [86].

EpCAM-specific monoclonal antibodies that target CD326 are highly expressed in hepatic tumor cells, such as hepatoblastoma. Therefore, for hepatoblastoma patients that have a high chance of relapse post liver transplantation surgery following hematopoietic stem cell transplantation, the EpCAM-specific antibody can be used as an optional therapy for hepatoblastoma [111].

##### Toxin-Based Immunotherapy

Cancer immunotherapy aims to enhance the patient’s immune response against cancer cells. In a similar manner, toxin-based immunotherapy utilizes toxin conjugated with cancer cell surface protein-binding molecules to create toxin conjugate. Toxin-based immunotherapy has been used in the treatment of recurrent glioblastoma multiforme. MDNA55 is a toxin-based immunotherapy that targets the interleukin-4 receptor with a 12-month OS rate of 55%, showing promising survival outcomes. It could be a potential therapeutic option for patients with recurrent glioblastoma multiforme when it is administered at a high dose, regardless of the patient’s interleukin-4 receptor expression levels [83].

##### Adoptive Cell Therapy

Adoptive cell therapy is a type of cancer immunotherapy that modifies immune cells, such as T-cells, to enhance their anti-cancer activity by adding a Chimeric Antigen Receptor (CAR), generating CAR T-cells. Epidermal growth factor receptor variant III (EGFRvIII) is a common mutation found in 30% of glioblastoma multiforme cases; thus, treatment with CART-EGFRvIII is shown to be safe without EGFR-directed toxicity, and treated patients have a median OS of 8 months [82].

Tisagenlecleucel (CTL019) is another CAR T-cell therapy mainly used in acute lymphoblastic leukemia (ALL), which targets CD19-expressing cancer cells. Tisagenlecleucel treatment has led to a 3-year OS rate of 63% in B-cell ALL [87], an 88% complete remission rate in down syndrome-associated ALL, an 81% overall remission rate at 3 months, and a 90% OS rate at 6 months [112]. However, 77% of patients treated with tisagenlecleucel experience Grade 3 or 4 adverse effects, which include cytokine-release syndrome, infections, prolonged cytopenias, neurologic events, and tumor lysis syndrome. Despite the adverse effect, the patient’s quality of life has improved in three months since 81% of patients treated with tisagenlecleucel achieved a minimally clinically important difference when measured by the Pediatric Quality of Life Inventory and 67% when determined by the European Quality of Life-5 Dimensions questionnaire.

##### Oncolytic Viruses

An oncolytic virus is a genetically modified virus that targets cancer cells for oncolysis and is also used to induce an immune-mediated anti-tumoral response. In recurrent glioblastoma, oncolytic DNX-2401 virotherapy plus the PD-1 inhibitor pembrolizumab has shown a beneficial effect with a 12-month OS of 52.7% [95]. In neural stem cell virotherapy, NSC-delivered CRAd-Survivin-pk7 (NSC-CRAd-S-pk7), a neural stem cell used for treating gliomas, has been shown to be feasible and safe in lower doses [113]. In addition, multiple doses of this virotherapy are currently in Phase I trials for recurrent high-grade gliomas (NCT05139056).

#### 4.2.2. Targeted Therapy

Targeted therapy is a highly specific drug targeting particular molecules or pathways in cancer progression. Targeted therapy can be classified based on the inhibited target.

##### Tyrosine Kinase Inhibitors (TKIs)

TKIs have been extensively used as targeted therapy for rare cancer patients by altering tyrosine kinase activity to block phosphorylation on target proteins. Imatinib mesylate, the first-generation BCR-ABL-targeted TKI that also inhibits c-kit and platelet-derived growth factor receptor (PDGFR) kinases, has been studied for Kaposi’s sarcoma treatment, yet it only shows 33.3% of a partial response [90]. With the same target as imatinib, a second-generation TKI, nilotinib is generated for patients who do not respond to imatinib. In CML patients, nilotinib treatment has led to 57.6% of patients achieving or maintaining a major molecular response and 81.8% of patients having a complete cytogenetic response [114]. Moving on to the third generation of TKI, ponatinib has the same target as previous generations of TKI, but it has been enhanced by including the T315I mutation. As a result, it has shown beneficial effects after treatment with 45 mg of ponatinib as a starting dose, which leads to 44.1% of primary endpoint achievement in 12 months [115]. Other types of TKIs are also used in rare cancer treatments, including pazopanib and cabozantinib for Merkel cell carcinoma, sorafenib, which targets vascular endothelial growth factor receptor (VEGFR), PDGFR, and RAF kinases for thymic carcinoma, EGFR and VEGFR for esophageal cancer, ABL001 (Asciminib) [91], as well as bosutinib in CML [116].

##### Poly (ADP-Ribose) Polymerase (PARP) Inhibitors

PARP inhibitors work by blocking the PARP enzyme that is involved in the DNA repair process and, therefore, killing cancer cells, especially those with DNA repair defects, including BRCA1/2 mutations. Veliparib is a PARP inhibitor reported to significantly enhance the efficacy of temozolomide in glioblastoma with MGMT promoter hypermethylation [89]. Similar to veliparib, a combination of PARP inhibitors such as talazoparib, niraparib, olaparib, and veliparib can synergize with temozolomide or SN-38, which leads to effective chemo sensitization in Ewing sarcoma and promotes caspase-dependent cell death [117].

##### Proteasome Inhibitors

Proteasome inhibitors such as bortezomib target the ubiquitin-proteasome pathway and inhibit pro-apoptotic factor degradation. It has been used in T-ALL with a 4-year OS rate of 88.3% without excessive toxicity [92]. In sarcomas, the combination of BCL-2 inhibitor ABT-199 and bortezomib leads to higher apoptosis induction as bortezomib inhibits the anti-apoptotic protein MCL-1 due to BOK (BAX/BAK homologue) accumulation, resulting in apoptosis induction through several pathways [118]. Carfilzomib, a proteasome inhibitor that has been used in myelomas, was recently shown to induce cellular apoptosis and alter the cellular metabolism in esophageal cancer mediated by Activating Transcription Factor 3 (ATF3) that binds to lactate dehydrogenase A (LDHA) and suppresses its metabolic alteration upon carfilzomib treatment [119].

##### Epigenetic Therapy

Recently, epigenetic therapy has gained a lot of attention, along with the exploration of cancer epigenetic etiology. Epigenetics is referred to as a biology field that explores heritable epigenetic modifications without altering the DNA sequence [120]. There are three types of epigenetic regulation [121]. The first type is DNA chemical modification, such as DNA methylation, which leads to the development of DNA methyltransferase inhibitors as one of the key components of epigenetic drugs. The second type is the post-translational modification that leads to the generation of histone deacetylase inhibitors. The third type is the alteration of gene expression by noncoding RNA, leading to the synthesis of RNA-targeted therapies that modulate the RNA to influence the epigenetic landscape. Epigenetic drug studies have been conducted to gain a deeper understanding of the epigenic alterations in several rare cancers, including an investigation of the role of miR-145-5p in thymic epithelial tumor progression using HDAC inhibitor Valproic Acid treatment [122] and the role of epigenetics on Merkel cell polyomavirus (MCPyV) in Merkel cell carcinoma etiopathogenesis [123] on the efficiency of targeted drug delivery to cancer cells.

One of the examples of targeted drug delivery is paclitaxel, which is commonly used for the treatment of many types of cancer, including the rare cancer Kaposi’s sarcoma [74,124]. Therefore, drug delivery compounds such as albumin and liposomes are commonly used for delivering paclitaxel. Albumin is an abundant plasma protein that is used for the paclitaxel delivery system as it has long blood retention and high drug-binding capacity; thus, the albumin-bound paclitaxel, also known as nab-paclitaxel, is used to increase the solubility of hydrophobic paclitaxel [125]. Another targeted drug delivery system, liposomal paclitaxel, uses the liposome to protect the paclitaxel from degradation due to abiotic factors that can affect its stability, such as temperature, humidity, and oxygen [126]. Liposomes also ensure smooth delivery of paclitaxel to the target site while decreasing the systemic toxicity of the encapsulated paclitaxel [126]. A study combined both of these drug delivery systems by encapsulating the albumin-bound paclitaxel with the liposome, resulting in increased antitumor effects and drug accumulation at the tumor site through enhanced permeability retention and endocytosis of the liposome [127]. The mechanism of this drug system is shown in Figure 2.

#### 4.2.3. Transplantation

In terms of cancer modality options, transplantation is a therapy that is only available for certain types of rare cancers.

Liver transplantation is an alternative for patients with extensive and unresectable hepatoblastoma, which is found in approximately 60% of patients on the first diagnosis. Although ideally chemotherapy is used as a neoadjuvant treatment before the transplantation, the tumor may remain unresectable, and approximately 15% of patients with unresectable hepatoblastoma still require liver transplantation [128].

Blood and marrow transplantation (BMT) has become one of the most common transplant therapies for rare blood cancers such as CML and ALL. BMT refers to the replacement of leukemic cells in bone marrow with healthy cells. BMT consists of allogeneic stem cell transplantation, in which the healthy transplant cells come from a compatible donor, and autologous stem cell transplantation, in which the healthy transplant cells come from the patient themselves. The use of allogeneic hematopoietic stem cells (allo-HSCTs) in patients with T-ALL shows a notably higher 10-year OS rate, with 40% of the OS rate for matched related donors HSCT vs. 45% for allo-HSCT vs. 26% for chemotherapy alone [128].

In Ewing sarcoma, treatment using high-dose busulfan and melphalan chemotherapy along with autologous stem-cell transplantation improves event-free survival (EFS) compared to the standard chemotherapy consisting of vincristine, dactinomycin, and ifosfamide (3-year EFS 69% vs. 56.7% and 8-year EFS 60.7% vs. 47.1%) and OS (3-year OS 78.0% vs. 72.2% and 8-year OS 64.5% vs. 55.6%) [93].

Although allo-HSCT is a promising therapeutic option for rare blood cancers, the Graft-versus-Host Disease (GvHD) that happens following the transplantation often hampers the success of an otherwise potentially curative transplant, leading to severe complications and even death. Commonly used metabolites for GVHD prevention are combinations of calcineurin inhibitors, which include methotrexate, cyclosporine, and tacrolimus. Another type, such as mycophenolate mofetil, an inosine monophosphate dehydrogenase inhibitor, is also being studied for its GVHD prophylaxis effect, yet the combination of Tacrolimus and methotrexate has been shown to be the golden standard that leads to a better outcome compared to the cyclosporine and mycophenolate mofetil combination [128].

#### 4.2.4. Drug Combination

Drug combination therapy is a type of cancer therapy that involves merging several drugs of different treatment types to achieve an increase in drug efficacy, overcome drug resistance, and decrease the treatment duration. The combination can come from the same group of drugs with the same target or combine different types of cancer modalities.

A Phase 2 clinical trial study showed that combined treatment of PD-1 inhibitornivolumab (240 mg, q2 weeks) and CTLA-4 inhibitor ipilimumab (1 mg/kg; q6 weeks) in Merkel cell carcinoma patients led to 31% of subjects achieving an objective response and a 15% complete response. The addition of SBRT (24 Gy; week 2) did not enhance the efficacy of the combined treatment, but the combination of nivolumab and ipilimumab can be used as a first-line therapeutic option for Merkel cell carcinoma [70].

Carboplatin, a chemotherapeutic drug that binds with DNA to form DNA adducts, is combined with the plant alkaloid paclitaxel and has led to 7.5 months of PFS in thymic carcinoma patients [94]. Carboplatin is also used along with amrubicin, an anthracycline chemotherapeutic drug that inhibits topoisomerase II. The combination of carboplatin and amrubicin as first-line therapy has led to 42% ORR in thymic carcinoma with a 7.6-month PFS. Both studies have reported that over 80% of patients experienced transient and manageable Grade 3 or 4 hematological toxicities such as neutropenia [94].

In a previous study, a combination of 60 Gy radiotherapy, temozolomide, and bevacizumab treatment prolonged the PFS but did not improve the OS in glioblastoma patients. ABT-414 (depatuxizumab mandolin) combined with temozolomide is tested for its effect in glioblastoma patients and has shown 25.25% of 6-month PFS and 69.1% of 6-month OS, and it has been shown to exhibit a well-tolerated safety profile and satisfactory pharmacokinetic characteristics in glioblastoma [96]. However, the Phase 3 clinical trial of temozolomide combined with nivolumab did not increase the PFS (10.6 months vs. 10.3 months) or OS (28.9 months vs. 31.2 months) compared to the control [97]. An oncolytic virus is also used as an upfront treatment for glioblastoma patients, showing that the neural stem cell line that is treated with an oncolytic virus before radiation and chemotherapy leads to higher cellular apoptosis and cytotoxicity [98]. Combination therapy of cisplatin, 5-fluorouracil, vincristine, and doxorubicin (C5VD) showed a significant effect in children with hepatoblastoma with a 95% OS rate and a 95% event-free rate (5 years) [99].

An Ewing sarcoma treatment using a combination of CDK4/6 and IGF1R inhibitors has revealed that CDK4/6 and IGF1R have a synergistic effect that leads to cell cycle and PI3K/mTOR signaling repression [100]. All 20 Kaposi’s sarcoma patients survived in a Phase 2 study that applied Valganciclovir before the initiation of combined antiretroviral therapy (cART), while treatment with cART alone showed a 15% mortality rate (NCT03296553). Meanwhile, in esophageal cancer patients with PD-L1 expression of 1% or greater, nivolumab and ipilimumab combined with chemotherapy increased the OS compared to chemotherapy alone (12.7 vs. 10.7 months) [102].

TKI combined with nilotinib and ruxolitinib, which inhibit JAK2 and TYK2 in the JAK-STAT pathway in CML patients, has shown that the combination is safe, and 4 out of 10 patients have undetectable BCR-ABL1 transcripts [103].

VAY736, an anti-B-cell activating factor receptor, combined with EW-7197, a TGF-β receptor 1 inhibitor, led to a significant reduction in ALL cell growth in the bone marrow, spleen, and blood in the in vivo study [104]. In unresectable/metastatic anal cancer, a combination of 403MO Atezolizumab with bevacizumab led to a 11.6-month OS rate, and out of all the 20 patients, 2 had a partial response and 11 had stable disease [72].

### 4.3. Clinical Trials

There are a number of completed and ongoing clinical trials to test the new treatment efficacy in rare cancers, including the abovementioned rare cancer modalities. These studies are summarized in Table 5.

### 4.4. Novel Advances in Other Types of Rare Cancer Treatment

The distinction between the treatment of rare cancers and common cancers needs careful consideration due to differences in various cancer types, the absence of standardized treatment regimens, and the lack of targeted therapies. Among 230 types of rare cancers, only a few have made significant advancements in finding novel therapies. Apart from the examples mentioned above, there are also some advances in the treatment of other types of rare cancers.

In neuroendocrine tumor (NET), the use of dose-escalated peptide receptor radionuclide therapy (PRRT), which allows targeted radioactivity to the tumor by somatostatin receptor (SSTR), is shown to be effective for metastasized NET treatment with relatively low systemic toxicity. As it targets SSTR, the selection of patients with unresectable advanced disease for this treatment depends on the patient’s SSTR by using somatostatin receptor-positron emission tomography. Compared to other treatments, the rapidly growing adoption of PRRT is attributed to several factors, including the specifically targeted treatment, the low toxicity profile at low doses, and the convenient PRRT administration schedule with 1-day cycle completion at a 10–12-week interval [131]. Despite its potential, the current use of PRRT is limited to palliative treatment with a low dosage, as the high dose of the PRRT regimen for tumor remission can lead to systemic toxicity. Therefore, the PRRT combination with the β-particle emitter ^177^Lu-octreotate-based PRRT has been a standard treatment for gastroenteric NETs that express SSTR2 and SSTR5 [132]. Other potential radioisotopes, Tb-161 and α-emitter Ac-225, are also being developed for NET Grade 1 to Grade 3 therapy [132]. In addition, a high dose of a combination of bifunctional metal chelator (DOTAM) and SSTR-targeting peptide (TATE), 212Pb-DOTAMTATE, has been shown to be well tolerated, with an 80% radiologic response at a dose of 2.50 MBq/kg/cycle.

Another relatively new advancement in rare cancer therapy is the use of trastuzumab deruxtecan, a Human Epidermal Growth Factor Receptor 2 (HER2)-targeting antibody–drug conjugate and topoisomerase I inhibitor payload for the treatment of uterine serous carcinomas (USC), as USC is one of the rare cancers known to overexpress HER2/neu [99]. The study showed that six cycles of trastuzumab increased the median PFS (12.6 months) compared to control (8.0 months) [133]. A follow-up study by the same author showed that trastuzumab yielded the greatest benefit in Stage III and Stage IV HER2/Neu-positive USC with trastuzumab as the primary treatment, as it led to a higher median PFS (17.7 months) compared to control (9.3 months) and a higher OS (29.6 months vs. 24.4 months) with similar toxicity observed in both groups. Although trastuzumab is known to target HER2, a recent study showed that trastuzumab deruxtecan resulted in good efficacy in both HER2-high and HER-2-low UCS patients with an ORR of 54.5% vs. 70.0%, a median PFS of 6.2 months vs. 6.7 months, and an OS of 13.3 months vs. not reached [134].

## 5. Challenges and Perspectives

Therapy for rare cancer is a complex process with a unique set of challenges, mainly due to the scarcity of data, limited research, and delayed diagnosis. The challenges/limitations and future perspectives of rare cancer diagnosis and therapy are summarized in Table 3.

The small number of clinical sample populations becomes the greatest challenge faced in the development of rare cancer treatments. Clinical trials are terminated due to low participant enrolment. Although this problem might be resolved by intercontinental clinical trial collaboration, the absence of a standardized treatment protocol will lead to a lack of homogeneity in the sample. Together with the inadequate number of ongoing clinical trials due to a lack of accrual, this becomes a greater challenge in the rare cancer research field. An example of intercontinental collaboration is the International Berlin–Frankfurt–Münster Study Group in 2002–2007 across 15 countries on three continents to study the impact of delayed intensification treatment of childhood ALL [135]. Although there is no significant condition improvement achieved by more intense or prolonged delayed intensification, this clinical trial has shown a great example of intercontinental collaboration in rare cancer treatment development. Another example is the RARECARENet, a rare cancer database collected from 94 regions across seven countries that are members of the European Union [136].

A low sample population also leads to limited research and clinical data that can be used to develop novel targeted therapies. Moreover, with the broad heterogeneity spectrum of rare cancers, the research has to be conducted separately and specifically on each type of rare cancer and therapy, making the research of rare cancers even more challenging. The limited existing research on rare cancers is the precise reason for the compelling need to actively promote further research in rare cancer therapies. Enhancing our understanding of rare cancers is imperative to the advancement and emerge of novel therapeutic modalities.

A rare cancer diagnosis is especially challenging due to its rarity; it is prone to misdiagnosis and delayed diagnosis until it reaches an advanced stage. A delayed diagnosis can limit the available therapy options compared to when the cancer is detected in its early stages.

Since developing drugs for rare cancers can be an economically challenging for drug companies due to the small population of the potential market, not many companies are inclined to invest in the drug development for rare cancer. Therefore, the U.S. Food and Drug Administration pioneered the Orphan Drug Act, an act used to encourage high-cost drug development tailored for rare diseases, including rare cancers.

To overcome these challenges in rare cancer therapy, all parties, including healthcare professionals, researchers, and organizations, are required to collaborate and enhance the quality of care and drug development for rare cancers.

## 6. Conclusions

Rare cancers are low-incidence cancers that comprise a wide range of malignancies with their own characteristics and different approaches to diagnosis and treatment regimens. The small number of clinical populations poses a formidable challenge in the development of rare cancer diagnostic tools and the expansion of therapeutic options. Although conventional diagnosis and treatment might be effective in common cancer, their application exhibits limitations in rare cancer. Therefore, novel diagnostic strategies and therapy regimens tailored specifically for rare cancers are necessary. In this review, we summarized the advances of modern diagnostic tools as well as the therapeutic modalities for rare cancers. Challenges, including rare cancer drug development, may be untangled by the global collaboration of all contributing parties in enhancing therapy regimens for rare cancers.

## Figures and Tables

**Figure 1 ijms-25-01201-f001:**
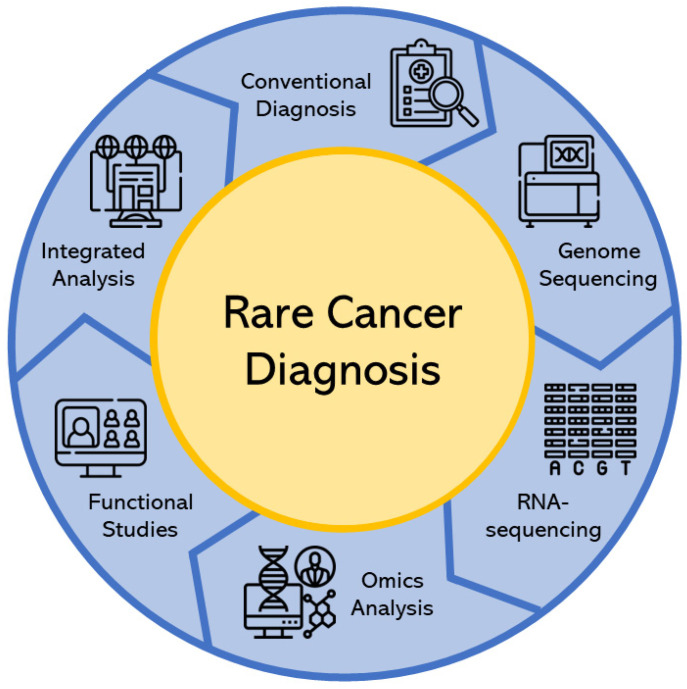
Procedure flow of modern diagnosis of rare cancers. Conventional diagnosis will be performed at first and inconclusive result will lead to more intricate diagnosis such as genome sequencing and RNA-sequencing which data can be used for omics analysis if needed. All data from previous diagnosis and research will be analyzed further to determine linked data and correlations in functional studies. These studies will provide a large amount of data which can be used in integrated analysis to improve diagnosis methods and prognosis.

**Figure 2 ijms-25-01201-f002:**
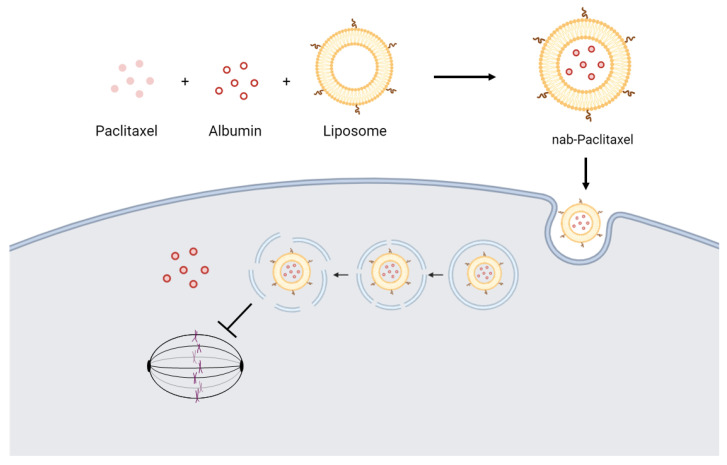
Mechanism of albumin-bound paclitaxel-encapsulated liposome delivery system. Paclitaxel is bound with albumin and encapsulated within liposome. Liposome is internalized by tumor cells by endocytosis. Liposome can release paclitaxel after the endosome lysis. The paclitaxel is released into the cytoplasm, stabilizing microtubules and inhibiting mitosis.

**Table 1 ijms-25-01201-t001:** Advantages and limitations of conventional cancer diagnostic methods.

Diagnostic Method	Description	Advantages	Limitations	Ref.
DNA Microarray Analysis of Tumors	Facilitates the simultaneous analysis of the expression levels of numerous genes. Hybridization of labeled genetic material from a sample with a probe allows simultaneous expression level assessment of thousands of genes.	Allows for the simultaneous examination of thousands of genes. Facilitates the molecular subtyping of tumors based on gene expression patterns. Genetic changes indicative of cancer and potential biomarkers of cancer can be detected at early stages, allowing for early diagnosis and intervention.	Requires sophisticated bioinformatics tools for interpretation. Expensive. Technical variations and noise in microarray data may generate misleading results. Obtainment of high-quality tumor tissue poses a major challenge.	[10]
In Situ Hybridization	Identifies specific nucleic acid sequences. Involves the hybridization of a complementary DNA probe to the target sequence and the evaluation of various recognized aberrations, including rearrangements resulting from translocations, insertions, inversions, deletions, and amplification.	Allows the visualization and localization of specific DNA or RNA sequences within cells and tissues. Can be performed on intact cells or tissue sections, which is essential for understanding the localization of genetic alterations within the complex architecture of tissues affected by cancer. Can detect various genetic aberrations. Can specifically identify RNA molecules, providing insights into gene expression patterns and potential biomarkers.	The sensitivity and specificity are very dependent on technicalities. Primarily a qualitative technique, and quantitative analysis may be challenging. Requires a high-quality tissue sample. Each ISH assay targets a limited amount of a specific nucleic acid sequence.	[11]
Flow Cytometry	Utilizes lasers for cell counting, sorting, and biomarker detection. Allows simultaneous analysis of physical and chemical characteristics in thousands of particles per second. Characterizes cancer cells by their physical and fluorescent properties to measure proteins expressed by specific immune cell subpopulations.	Enables lineage definition and differentiation state of cell populations. Enables the evaluation of cell proliferation, DNA ploidy analysis, and the identification of rare events in a short time. Able to analyze cells faster with relatively low sample volume requirements and costs, as well as shorter and easier sample preparation and instrument set-up protocols.	Does not provide spatial information about the cells within tissues. May face difficulties when analyzing cells with similar sizes or when studying highly complex cell populations. High-quality flow cytometry instruments can be expensive. The availability of fluorochromes for labeling is limited and scarce; however, its usage is important and can impact the accuracy of multicolor analysis. Multivariate data generated by flow cytometry requires sophisticated analysis tools.	[12]
Tumor Markers	Detection of tumor-related substances that aid in distinguishing a tumor from normal tissue or confirming the presence of a tumor. Tumor markers include cell surface antigens, cytoplasmic proteins, enzymes, hormones, antigens produced by tumor and fetal tissues, receptors, oncogenes, and their products.	Capable of diagnosing the origin of cancer even in patients with advanced widespread disease. Tumor marker levels may provide insights into the extent of cancer progression. Assists in determining the prognosis by indicating the likely speed of cancer progression based on the stage of the disease.	Exhibits very low concentrations in tissues with small, early-stage cancer lesions. Both normal and cancer cells can produce most tumor markers, contributing to challenges in specificity. Proteins or modified proteins associated with tumor markers may vary in each patient, which makes accurate interpretation challenging. Individuals with cancer may not always exhibit elevated tumor marker levels in their blood, and even then, these markers lack the specificity needed to definitively confirm the presence of cancer.	[13]
Biopsy	Taking a small sample of cancer tissue so it can be examined under a microscope.	Provides a definitive diagnosis by direct visualization. Allows for genetic and molecular analyses	Invasive, risk of bleeding or infection Possibility of sampling error. Recovery time varies. Challenging for tumors located in inaccessible or sensitive areas.	[7]
Pressure Application	Gentle pressure is applied to various body parts to observe abnormalities inside the body.	Helpful for basic observation to identify the presence of cancer in a qualitative manner.	The accuracy and precision depend entirely on the examiner’s skill. Cannot identify cancers deep inside the body.	[14]
Fine Needle Aspiration Cytology (FNAC) and Core Needle Biopsy (CNB)	FNAC extracts a small amount of cancer cells from the suspicious lesion on the body by using a small needle to examine it under a microscope. Meanwhile, CNB has a larger needle, so it can obtain a small tissue.	FNAC is minimally invasive. Results can be produced quickly. FNAC is a cost-effective procedure. CNB provides detailed information about the tumor’s structure, architecture, and surrounding tissues. Allowing for a more accurate diagnosis. CNB is effective in obtaining samples from deeper lesions within the body.	May not provide enough tissue for a definitive diagnosis. FNAC may not provide detailed information about the tumor structure and architecture. The accuracy of FNAC is sensitive to human error. CNB is invasive, involves a large needle, and has a high risk of complications. Risk of bleeding at the biopsy site. Recovery time after CNB may be longer compared to FNAC due to the larger incision or puncture.	[14]
Histochemistry and Cytochemistry	Involves chemical staining of tissues (histochemistry) and individual cells (cytochemistry) to identify and observe the distribution and abundance of specific molecules associated with cancer development.	Allow for specific staining of cellular components. Help identify specific biomarkers or molecular features associated with cancer cells. Provide information about the tissue architecture. Can localize proteins within cells, revealing the subcellular distribution of key molecules relevant to cancer development.	Interpretation of staining patterns can be subjective. Obtaining quantitative data can be challenging. May not capture heterogeneity at the single-cell level within a tumor. The fixation process can disturb the native state of cellular components and result in inaccuracy. Time-consuming.	[14]
Electron Microscopy	Physical examination of cancer by using an advanced microscope to produce a high-quality and detailed image of the cancer surface.	Provides extremely high-resolution results, which provide insights into the specific alterations that occur in cancer cells. Helps identify subcellular abnormalities, which can be indicative of cancer. Allows for a detailed morphological analysis of cellular components.	Complex and expensive. Preparation of samples can be challenging as it involves intricate procedures. Time-consuming and may not be suitable for rapid diagnosis. Not suitable for studying live cells. Result lacks functional data such as specific molecular interactions inside the tumor. Does not provide molecular information about the composition of cellular structures.	[14]
Immuno-histochemistry (IHC)	Utilizes antibodies to detect the presence or absence of biomarkers associated with various types of cancer, allowing for their visualization under a microscope.	Allows for the visualization and quantification of protein expression levels within cancer cells. Provides information about the tissue morphology. Highly sensitive and able to detect low levels of protein expression.	Interpretation of IHC results can be subjective. Restricted to the detection of proteins for which specific antibodies are available. Sensitive to technical issues. Intra-tumor heterogeneity may result in variations in protein expression levels, which affect accuracy. Standardization of IHC protocols is challenging.	[14]
Histology	Microscopic examination of suspected tissue that may have cancer inside it that has been excised by biopsy or surgical resection.	Provides direct visual observation of tissue samples. Enables the assessment of tissue architecture. Provides a definitive confirmation by visually confirming the presence of cancerous cells.	Obtaining tissue samples often involves biopsies or surgical excisions, which pose risks to the patient and are not always feasible. Small or insufficient samples may lead to sampling bias. Requires a lot of time. Interpretation of histological slides can be subjective.	[15]
X-ray, Ultrasound, and Other Imaging Techniques	Methods of screening the body for abnormal tissue growth by using X-ray imaging or ultrasound.	Can detect the early stages of tumors and abnormalities, allowing for timely intervention and treatment. Generally non-invasive. Certain imaging modalities enable the visualization of the entire body. Ultrasound provides real-time imaging. Some imaging methods allow for the measurement of tumor size, growth rates, and response to treatment.	May have reduced sensitivity for detecting small lesions. Interpretation errors can arise due to overlapping structures or benign conditions mimicking cancer. X-ray and computed tomography (CT) scans with frequent exposure can pose health risks. Some advanced imaging techniques can be expensive. Often cannot offer detailed information about tissue composition and molecular features.	[15]

**Table 2 ijms-25-01201-t002:** Tools and technologies currently used in rare cancer diagnosis.

Diagnosis Technology	Tools	Examples
Genome sequencing	DNA-seq	Whole genome or exome sequencing analysis for detecting and diagnosing ovarian immature teratomas (University of California, San Francisco Medical Center) [51], squamous cell carcinoma of the prostate (Qingdao Municipal Hospital) [52]
	New generation sequencing	
	Third-generation sequencing	
RNA-sequencing	Single-cell RNA-seq	RNA-seq diagnosing urothelial cancer (Sandra and Edward Meyer Cancer Center) [53] and unclassified T cell lymphoma subtype (The First People’s Hospital of Yunnan Province) [54]
	Massively parallel single-cell RNA-seq	
Omics	Genomics	Whole genome or exome sequencing analysis diagnosing ovarian immature teratomas (University of California, San Francisco Medical Center) [51], squamous cell carcinoma of the prostate (Qingdao Municipal Hospital) [52] Multi-omics analysis to diagnose and identify hepatocellular carcinoma (University Hospital Basel) [55] and pan-cancer [56]
	Transcriptomics	
	Proteomics	
	Metabolomics	
	Integrated omics analysis	
	Liquid biopsies	
	Artificial intelligence and machine learning	
Functional studies	Immunohistochemistry	Immunohistochemistry diagnosing several patients with suspected squamous cell carcinoma of the prostate (Keio University Hospital) [57], ovarian immature teratomas (University of California, San Francisco Medical Center) [51], and malignant transformation of mature cystic teratoma [58] Flow cytometry used to conduct early diagnosis of patients suspected with myeloid sarcoma (Armed Forces Medical College) [59], lung carcinoid (Precura Center) [60], and diffuse large B-cell lymphoma (Bach Mai Hospital) [61] FISH technique for diagnosis of angiomatoid fibrous histiocytoma (Saitama Medical Center) [62] and renal cell carcinoma (Breach Candy Hospital Trust) [63], and other analysis that when combined can further improve the understanding of rare cancer etiology and diagnosis.
	Flow cytometry	
	Fluorescence in situ hybridization	
	Functional magnetic resonance imaging (fMRI)	
	Positron emission tomography (PET)	
	Cell culture studies	
	Functional genomics	
Integrated analysis	Genomic analysis Next-generation sequencing Comparative genomic hybridization	Integrated analysis diagnosed ovarian cancer (Harbin Medical University Cancer Hospital) [64] and colorectal cancer (The Second Affiliated Hospital of Zhejiang University Medical College) [65] Microarray analysis of tissues from patients undiagnosed with meningioma (Istituti Ospitalieri of Cremona) [66], synovial sarcoma (Memorial Sloan-Kettering Cancer Center) [67], and vestibular schwannoma (La Paz University Hospital) [68] Myeloid sarcoma (Armed Forces Medical College) [59], lung carcinoid (Precura Center) [60], and diffuse large B-cell lymphoma (Bach Mai Hospital) were diagnosed by flow cytometry [61] MRI, MR, and computed tomography were used for the visual examination of patients with undiagnosed basaloid carcinoma of the prostate (Royal Marsden Hospital) [69], malignant transformation of mature cystic teratoma (Nepal Cancer Hospital and Research Center) [58], squamous cell carcinoma of the prostate (Keio University Hospital and Qingdao Municipal Hospital) [52,57] Genome sequencing analysis diagnosed and detected cancerous sequences in patients with ovarian immature teratomas (University of California, San Francisco Medical Center) [51] and squamous cell carcinoma of the prostate (Qingdao Municipal Hospital) [52]
	Transcriptomic analysis Microarray analysis RNA-sequencing	
	Proteomic analysis Mass spectrometry Protein microarrays	
	Epigenomic analysis DNA methylation profiling Histone modification analysis	
	Imaging techniques Medical imaging (MRI, CT, PET, etc.) Radiomics	
	Clinical and pathological data Electronic health records (HER) Pathological evaluation	
	Machine learning and bioinformatics	

**Table 3 ijms-25-01201-t003:** Challenges and perspectives of rare cancer therapies.

Challenges	Impact	Perspective
Low incidence	Small number of clinical samples	Intercontinental clinical trial collaboration for data collection
	Termination of clinical trials due to a lack of participants	
Lack of standardized protocol	Lack of homogeneity in the sample	Collaboration of healthcare professionals, researchers, and organizations in establishing international guidelines for rare cancer treatment
Delayed diagnosis due to rarity	Limited therapeutic options as the cancer advanced	The use of novel diagnosis tools and artificial intelligence integration in rare cancer diagnosis
Economic challenge in developing rare cancer drugs	Inadequate number of ongoing clinical trials	Orphan Drug Act pioneered by the U.S. Food and Drug Administration

**Table 4 ijms-25-01201-t004:** Rare Cancer therapeutic modalities.

Treatments	Rare Cancer Types									
	Merkel Cell Carcinoma	Thymic Carcinoma	Glioblastoma Multiforme	Hepato-Blastoma	Ewing Sarcoma	Kaposi’s Sarcoma	Esophageal Cancer	Chronic Myeloid Leukemia	Acute Lymphoblastic Leukemia	Anal Cancer
Surgery	Mohs micrographic surgery	VATS thymectomy and Robotic VATS	Fluorescence-Guided Surgery	Robot-assisted hepatectomy	Rotationplasty	Cryosurgery	Minimally invasive esophagectomy	Not Applicable	Not Applicable	Abdomino-perineal Resection (APR)
Radiation Therapy	Stereotactic body radiation therapy (SBRT) [70]	SBRT	Image-Guided Radiation Therapy	Not Applicable	Proton Therapy	Electron Beam Radiation Therapy	Chemo-radiation	Not Applicable	Not Applicable	Chemo-radiation Radio-frequency Ablation
Chemotherapy	Platinum + etoposide [71]	Alimta (Peme-trexed) [72]	Metronomic Temozolomide	Cisplatin, 5-fluorouracil, and Vincristine [73]	Nab-paclitaxel [74]	Liposomal Chemotherapy		Platin and Fluoro-pyrimidine	Daunorubicin [75] 5-Azacytidine [76]	Fludarabine, Cyclophos-phamide
Immune Therapy	Poly-ICLC + Tremeli-mumab + Durvalumab [77]	PD-1/PD-L1 inhibitor [78]	Nivolumab [79] Cetuximab and Bevacizumab [80] ABT-414 [81] EGFRvIII CAR T Cells [82] MDNA55 [83]	EpCAM-specific monoclonal antibodies	Vigil [84]	Pembro-lizumab [85]	Pembro-lizumab [79]	IFN-α	Blinatumomab [86] CTL019 (Tisagenlecleucel) [87]	Retifanlimab (INCMGA00012) [88] Axalimogene Filolisbac (ADXS11-001) [50]
Targeted Therapy	Pazopanib + cabozantinib	Sorafenib	Veliparib [89]	Cabozantinib-S-Malate	Talazoparib, niraparib, olaparib, veliparib	Imatinib Mesylate [90]	EGFR VEGFR targeting agents	ABL001 [91]	Bortezomib [92]	Not Applicable
Transplant	Not Applicable	Not Applicable	Not Applicable	Liver transplantation	Autologous Stem Cell Transplantation [93]	Not Applicable	Not Applicable	Allogeneic Hematopoietic Stem Cell Transplantation	Allogeneic Hematopoietic Stem Cell Transplantation	Not Applicable
Combined Therapy	Nivolumab + Ipilimumab + SBRT [70]	Carboplatin + paclitaxel [94] Carboplatin + amrubicin	Oncolytic DNX-2401 virotherapy + pembrolizumab [95] ABT-414 + Temozolomide vs. Lomustine [96] Temozolomide + Radiation + Nivolumab [97] Oncolytic adenovirus + radiation + chemotherapy [98]	Cisplatin/5FU/ Vincristine [99]	CDK4/6 and IGF1R Inhibitor [100]	Valganciclovir and combined Antirretroviral Therapy (cART) [101]	Nivolumab + chemotherapy, nivolumab + ipilimumab [102]	Ruxolitinib + Tyrosine Kinase Inhibitors [103]	VAY736 antibody + EW-7197 [104]	403MO Atezolizumab + bevacizumab [72]

**Table 5 ijms-25-01201-t005:** Clinical trials on rare cancer interventions (ClinicalTrials.gov, accessed 15 October 2023).

Cancer Type	NCT Number	Phase	Treatment Arms	Ref.
Merkel cell carcinoma	NCT03071406	Phase 2	Nivolumab + Ipilimumab with or without SBRT	[105]
	NCT02643303	Phase 1 Phase 2	Tremelimumab + IV Durvalumab + Poly-ICLC	[77]
Thymic carcinoma	NCT03921671	Phase 2	Ramucirumab + Carbo-Paclitaxel	[94]
	NCT00198133	Phase 2	Alimta (Pemetrexed)	[129]
Glioblastoma multiforme	NCT02017717	Phase 3	Nivolumab	[106]
	NCT01884740	Phase 1 Phase 2	Cetuximab + Bevacizumab	[80]
	NCT02573324	Phase 3	ABT-414 (Depatuxizumab mafodotin)	[81]
	NCT02858895	Phase 2	Convection-enhanced delivery of MDNA55	[83]
	NCT02798406	Phase 1 Phase 2	Oncolytic DNX-2401 virotherapy plus pembrolizumab	[95]
	NCT02152982	Phase 2 Phase 3	Temozolomide with or without Veliparib	[89]
	NCT02343406	Phase 2	ABT-414 Alone or ABT-414 + Temozolomide vs. Lomustine or Temozolomide	[96]
	NCT02667587	Phase 3	Temozolomide + radiation therapy with Nivolumab	[97]
	NCT03072134	Phase 1	Neural stem cell-based virotherapy of newly diagnosed malignant glioma	[98]
Hepatoblastoma	NCT02867592	Phase 2	Cabozantinib-S-Malate	[130]
	NCT03698994	Phase 2	Ulixertinib	
	NCT03220035	Phase 2	Vemurafenib	
	NCT03213665	Phase 2	Tazemetostat	
	NCT00980460	Phase 3	Cisplatin/5FU/vincristine	[99]
Ewing sarcoma	NCT01962103	Phase 1 Phase 2	Nab-paclitaxel	[74]
	NCT02511132	Phase 2	Vigil	[84]
Kaposi’s sarcoma	NCT00090987	Phase 2	Imatinib Mesylate	[90]
	NCT03296553	Phase 2	Valganciclovir + cART	[101]
Esophageal Cancer	NCT03189719	Phase 3	Chemo + Pembrolizumab	[79]
	NCT03143153	Phase 3	Nivolumab + chemotherapy, nivolumab + the monoclonal antibody ipilimumab	[102]
Chronic Myeloid Leukemia	NCT01844765	Phase 2	Nilotinib	[114]
	NCT02467270	Phase 2	Ponatinib	[115]
	NCT03106779	Phase 3	ABL001 or Bosutinib	[91]
	NCT03128411	Phase 2	Bosutinib	[116]
	NCT03610971	Phase 2	Ruxolitinib + Tyrosine Kinase Inhibitors	[103]
Childhood acute lymphoblastic leukemia	NCT04562792	Phase 2	Daunorubicin	[75]
	NCT01861002	Phase 1	5-Azacytidine	[76]
	NCT02101853	Phase 3	Blinatumomab	[86]
	NCT02435849	Phase 2	CTL019 (Tisagenlecleucel)	[87]
	NCT02112916	Phase 3	Combination chemotherapy with or without Bortezomib	[92]
Anal cancer	NCT03597295	Phase 2	INCMGA00012 following platinum-based chemotherapy (POD1UM-202)	[88]
	NCT02399813	Phase 2	Axalimogene Filolisbac (ADXS11-001)	[110]

## Data Availability

No new data were created or analyzed in this study. Data sharing is not applicable to this article.

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
