# Peer review of "An Overview of Advances in Rare Cancer Diagnosis and Treatment"

_ijms, 2024, doi:10.3390/ijms25021201_

Round 1

Reviewer 1 Report

Comments and Suggestions for Authors

Minor revision is needed.

Author Response

Replies to Reviewer #1

Major Comments

  1. Abstract should be re-written. Provide the novelty statement and the significance of

the need of the review. Provide the outline what the review is offering in terms of

novelty when compared with other existing literature.

Response: Thank you for the suggestion. We have re-written Abstract to provide the information requested.

  1. Provide clear statistics details rather than just mentioning them in few lines in para 2

of introduction (https://doi.org/10.17756/nwj.2023-122).

Response: As suggested, we have added clear statistics details in Introduction by giving several examples of the prevalence of rare cancers.

  1. Provide the list of limitations also.

Response: As suggested, we have added a new Table 5 in the Section 5 (Challenges and Perspectives) to list the limitations/challenges of rare cancer studies.

  1. Please provide the conclusion, regarding what final message authors are conveying to

the readers.

Response: Thank you for the suggestion. We have included Conclusion as Section 6 in the revised version to summarize the final messages we are conveying to the readers.

  1. Include more references. As the topic covered is extensive, having more systematic

discussion on the literature would be helpful for the readers.

Response: We had almost 200 references in our drafted manuscript but since there is a reference number limitation (no more than 100 references) for this journal, we had to reduce the reference number. As suggested, we added back some necessary references though the reference number could not exceed the limitation too much.

Minor Comments

  1. Provide the list of abbreviations in the beginning.

Response: Thank you for the suggestion. We have added a list of abbreviations in the beginning of the manuscript.

  1. Please limit the use of abbreviations. Use abbreviations if the intended word is

repeated more than thrice, otherwise state in full.

Response: Thank you for pointing it out. We have checked the use of abbreviations in the manuscript and revised the abbreviations based on the suggestion.

Reviewer 2 Report

Comments and Suggestions for Authors

1)      This manuscript requires an appropriate graphical abstract at the end of the introduction.

2)      I suggest the authors list the rare cancers in a table either in the main manuscript file or supplementary.

3)      The idea in Table 1 is good. However, the scheme of presentation is not good. The authors could change it to landscape and use few words instead of long sentences. E.g. Limitation of DNA microarray: expensive technology, well-trained personnel…

4)      One of the recent advances in cancer therapy is epigenetic therapy and epigenetic drug development since many if cancers have epigenetic etiology. The authors did not discuss it at al. they could use this recently published article.

https://www.mdpi.com/2073-4425/14/4/873

5)      Presentation of Table 3 is also not good. E.g the 3rd and 4th columns from the left are too long. They could convert to landscape and use smaller font sizes.

6)      The authors described genome sequencing, RNA sequencing, and omics analysis as modern technologies of rare cancers. My questions are (A) Which clinics/hospitals use these techniques to diagnose since they are expensive and time/cost-consuming? (B) The origin of approximately 50-70% of cancers is epigenetic and does not affect the genetic sequence.

7)      Some parts of the review are extremely short. E.g. targeted therapy (section 4.2.2) the authors explain the Ab-liposome and nanocarries prodrugs to explain it. They could use figures to explain the text further. Unfortunately, they did not use any figures. 

Author Response

Replies to Reviewer #2

1) This manuscript requires an appropriate graphical abstract at the end of the introduction.

Response: Thank you for the suggestion. We have added a graphical abstract at the end of the Introduction.

2) I suggest the authors list the rare cancers in a table either in the main manuscript file or supplementary.

Response: As suggested, we have included supplemental Table 1 to list the rare cancers.

3) The idea in Table 1 is good. However, the scheme of presentation is not good. The authors could change it to landscape and use few words instead of long sentences. E.g. Limitation of DNA microarray: expensive technology, well-trained personnel…

Response: Thank you for the suggestion. We have changed Table 1 to landscape and revised it extensively to make it more concise and clearer.

4) One of the recent advances in cancer therapy is epigenetic therapy and epigenetic drug development since many if cancers have epigenetic etiology. The authors did not discuss it at al. they could use this recently published article.

https://www.mdpi.com/2073-4425/14/4/873

Response: Thank you for the suggestion. We have included epigenetic therapy and epigenetic drug development in details in Section 4.2.2. Targeted Therapy.

5) Presentation of Table 3 is also not good. E.g the 3rd and 4th columns from the left are too long. They could convert to landscape and use smaller font sizes.

Response: Thank you for the suggestion. We have changed Table e to landscape and use a smaller font size.

6) The authors described genome sequencing, RNA sequencing, and omics analysis as modern technologies of rare cancers. My questions are (A) Which clinics/hospitals use these techniques to diagnose since they are expensive and time/cost-consuming? (B) The origin of approximately 50-70% of cancers is epigenetic and does not affect the genetic sequence.

Response: (A) We have added some examples of use these techniques to diagnose in the manuscript. The details are as below.

WGS can be utilized in clinical diagnosis of cancer or its rare variant. For example, Turro et al. successfully detected cancer-related mutations in peripheral blood from participants with personal history of tumor-associated phenotypes (identified through 11 Australian Familial Cancer Centers), which was used to measure their susceptibility of cancer and prediction of cancer type (PMID: 32581362).

Pei et al. reported accurate diagnosis by identifying disease-specific fusion genes in cancer patients from Department of Pathology at Fox Chase Cancer Center [29]. They successfully diagnosed an Ewing sarcoma by analyzing cancer tissue samples using RNA-seq analysis (PMID: 31232935).

(B) Since quite a lot of rare cancers are caused by gene sequence alterations (somatic or inherited) (see supplemental Table 1, cause of rare cancers), therefore, genome/ RNA/ omics analysis would help us in rare cancer diagnosis.

7) Some parts of the review are extremely short. E.g. targeted therapy (section 4.2.2) the authors explain the Ab-liposome and nanocarries prodrugs to explain it. They could use figures to explain the text further. Unfortunately, they did not use any figures.
Response: Thank you for the suggestion. We have expanded Section 4.2.2 (targeted therapy). We also included the details of albumin and liposome targeted drug delivery in the manuscript. Figure 2 was added to explain the mechanism of the drug delivery system.

Reviewer 3 Report

Comments and Suggestions for Authors

Due to their rarity, novel diagnostic techniques and treatments are undeveloped for most rare cancers. Furthermore, although the number of patients with each rare cancer disease is small, many patients are suffering from rare cancers because of the large number of rare cancers. This MS is a valuable review that focuses on rare cancers. Although the structure is good, the current MS needs to be improved for readability.

My comments for this MS are as follows.

1. The role of a review is to provide readers with a compact summary of findings in the field. On this point, Table 1 is redundant and difficult to understand. The authors should provide a more concise summary to help readers understand.

2. Regarding novel techniques for diagnosing rare cancers, I found the description comprised too many general introductions to each method and fewer descriptions for a rare cancer-specific perspective. In addition, a figure integrating the novel techniques will help readers understand the impact of diagnosing rare cancers.

3.        Regarding novel treatments: Among the many rare cancers, several diseases are described in this manuscript. It was difficult to understand why the authors listed these rare cancers among the many rare cancers.

4.        The authors stressed the importance of intercontinental collaboration for clinical trials. The description of successful examples will help readers understand.

Author Response

Replies to Reviewer #3

  1. The role of a review is to provide readers with a compact summary of findings in the field. On this point, Table 1 is redundant and difficult to understand. The authors should provide a more concise summary to help readers understand.

Response: Thank you for the suggestion. We have revised Table 1 extensively to make it more concise and clearer.

  1. Regarding novel techniques for diagnosing rare cancers, I found the description comprised too many general introductions to each method and fewer descriptions for a rare cancer-specific perspective. In addition, a figure integrating the novel techniques will help readers understand the impact of diagnosing rare cancers.

Response: As suggested, we have included some examples of rare cancer diagnosis using novel techniques in Section 3.2 (Modern Diagnosis) to give more descriptions for rare cancer-specific perspectives. Also, Figure 1 integrating the novel techniques in rare cancer diagnosis has also been provided.

  1. Regarding novel treatments: Among the many rare cancers, several diseases are described in this manuscript. It was difficult to understand why the authors listed these rare cancers among the many rare cancers.

Response: Several rare cancer types are described in the manuscript as they have recent advances in novel treatments. Other rare cancer types may not be included in the manuscript as they don’t have advances in treatment, or the novel treatments have not been applied in these diseases.

  1. The authors stressed the importance of intercontinental collaboration for clinical trials. The description of successful examples will help readers understand.

Response: As suggested, we have included some meaningful examples of intercontinental collaborations in Section 5 (Challenges and Perspectives). The details are as below.

An example of intercontinental collaboration had been done in 2002 to 2007 across 15 countries on three continents by the International Berlin-Frankfurt-Münster Study Group to study the impact of delayed intensification treatment of childhood ALL (PMID: 24344215). Although there is no significant condition improvement achieved by more intense or prolonged de-layed intensification, this clinical trial has shown a great example of intercontinental collaboration in rare cancer treatment development.

Another example is the RARECARENet, a rare cancer database collected from 94 region across seven countries that are the members of European Union (PMID: 28687376).

Round 2

Reviewer 2 Report

Comments and Suggestions for Authors

The authors provided my edition, however, they did not include a graphical (GA) abstract at the end of the introduction. I recommend publishing if they agree to add GA. 

Author Response

Replies to Reviewer #2

 The authors provided my edition, however, they did not include a graphical (GA) abstract at the end of the introduction. I recommend publishing if they agree to add GA.

Response: We have uploaded graphical abstract (separate high-resolution TIFF file) when we submitted our last revision. Something might go wrong in the system. We have contacted the editor and resubmitted the graphical abstract again. Please let the editor know if you still could not see it.

Reviewer 3 Report

Comments and Suggestions for Authors

Thank you for the revisions. My comments are as follows.

1)  3.1 conventional diagnosis section:

I am still doubtful about the necessity of this section. One reason for this is the appearance of conventional diagnosis methods such as IHC and flow cytometry in Table 2 “Modern tools ---. In addition, current table 2 is just a list of techniques. Authors are recommended to show encouraging examples of rare cancer diagnoses (described in the text) at each technique in this table. I believe this revision will help readers access the information regarding these successful examples in the field of diagnosis of rare cancers.

 2) Table5:

This is a good summary of current issues in the treatment of rare cancers. This should be placed first in the Treatment section.

3)I was unable to find Fig. 1 (Line 127) in the MS pdf.

Author Response

  1. 3.1 conventional diagnosis section:

I am still doubtful about the necessity of this section. One reason for this is the appearance of conventional diagnosis methods such as IHC and flow cytometry in Table 2 “Modern tools ---“. In addition, current table 2 is just a list of techniques. Authors are recommended to show encouraging examples of rare cancer diagnoses (described in the text) at each technique in this table. I believe this revision will help readers access the information regarding these successful examples in the field of diagnosis of rare cancers.

Response: Thank you for your suggestions. For an overview/summary of the rare cancer diagnosis, the conventional diagnosis section is a necessary part of this review. We do agree that Table 2 “previous title – modern tools and technologies in rare cancer diagnosis” has some overlapping techniques with Table 1. To avoid confusion, we have changed the title of Table 2 to “modern tools and technologies currently used in rare cancer diagnosis”. In addition, as suggested, we have included examples of rare cancer diagnosis for each technique in Table 2.

  1. Table 5:

This is a good summary of current issues in the treatment of rare cancers. This should be placed first in the Treatment section.

Response: Thank you for the suggestion. We have placed it first in the Treatment section (it is now as Table 3).

  1. I was unable to find Fig. 1 (Line 127) in the MS pdf.

Response: We have uploaded Fig.1 (separate high-resolution TIFF file, not incorporated in the MS PDF file) when we submitted our last revision. Something might go wrong in the system. We have contacted the editor and resubmitted the figures again. Please let the editor know if you still could not see it.